# Usefulness of the Novel Snare-over-the-Guidewire Method for Transpapillary Plastic Stent Replacement (with Video)

**DOI:** 10.3390/jcm10132858

**Published:** 2021-06-28

**Authors:** Akihiro Yoshida, Mamoru Takenaka, Kota Takashima, Hidekazu Tanaka, Ayana Okamoto, Tomohiro Yamazaki, Atsushi Nakai, Shunsuke Omoto, Kosuke Minaga, Ken Kamata, Kentaro Yamao, Yoriaki Komeda, Naoshi Nishida, Masatoshi Kudo

**Affiliations:** Department of Gastroenterology and Hepatology, Kindai University Faculty of Medicine, 377-2 Ohno-Higashi, Osaka-Sayama 589-8511, Japan; a.yoshida@med.kindai.ac.jp (A.Y.); g21001@edu.med.kindai.ac.jp (K.T.); atanakjp@gmail.com (H.T.); a-o-k@mail.goo.ne.jp (A.O.); chochiko.4kg@gmail.com (T.Y.); nakai_agmc@yahoo.co.jp (A.N.); shunsuke.oomoto@gmail.com (S.O.); kousukeminaga@yahoo.co.jp (K.M.); ky11@leto.eonet.ne.jp (K.K.); yamaken_volvo@yahoo.co.jp (K.Y.); y-komme@mvb.biglobe.ne.jp (Y.K.); naoshi@med.kindai.ac.jp (N.N.); m-kudo@med.kindai.ac.jp (M.K.)

**Keywords:** re-intervention, plastic stent, snare-over-the-guidewire

## Abstract

Unsuccessful stent replacement in transpapillary biliary drainage with plastic stents (PSs) has a significant impact on patient prognosis; thus, a safe and reliable replacement method is required. We aimed to compare the snare-over-the-guidewire (SOG) method, wherein the PS lumen is used as an access route to the biliary tract and the PS is removed with a snare inserted via the inserted guidewire, with the conventional side-of-stent (SOS) method, wherein the biliary approach is performed from the side of the PS. This retrospective single-center study included 244 consecutive patients who underwent biliary PS replacement between January 2018 and July 2020. The procedural success rates were compared between the two methods. A predictive analysis of unsuccessful PS replacement was also performed. The procedural success rate in the SOG group was significantly higher than that in the SOS group (*p* = 0.026). In the proximal biliary stenosis lesion, the same trend was observed (*p* = 0.025). Multivariate analysis also showed that the SOS method (*p* = 0.0038), the presence of proximal biliary stenosis (*p* < 0.0001), and parapapillary diverticulum (*p* = 0.0007) were predictors of unsuccessful PS replacement. The SOG method may be useful for biliary PS replacement, especially in cases of proximal hilar bile duct stenosis.

## 1. Introduction

Among the endoscopic retrograde cholangiopancreatography (ERCP)-related procedures, transpapillary drainage is one of the basic techniques routinely performed in a wide range of institutions, from high-volume centers to general hospitals [1,2,3,4,5,6]. Plastic stents (PSs), which can be used in both malignant and benign cases, are inexpensive, extensively used worldwide, and remain the gold standard for transpapillary biliary drainage 40 years after its introduction [7].

However, PSs have a shorter stent patency period than that of metallic stents and cannot be permanently inserted [8,9,10,11,12,13,14]. If stenosis is irreversible, endoscopic replacement is mandatory. In addition, recent advances in chemotherapy have prolonged the prognosis of patients with malignant biliary strictures, and the need for PS replacement has increased [15,16,17,18]. In other words, the procedure for biliary PS drainage should be designed with replacement in mind. Although PS replacement is a basic technique, it is sometimes challenging [19,20]. In cases of severe bile duct stenosis or flexion, the possibility of unsuccessful replacement is increased. Unsuccessful PS replacement may require a change in the drainage route, such as percutaneous transhepatic biliary drainage or endoscopic ultrasound-guided biliary drainage (EUS-BD), and may have a significant impact on the patient’s prognosis, especially in cases of malignant disease.

There is a need to establish a safe and reliable method of PS replacement.

In recent years, a PS replacement method using the characteristics of straight-type PS in EUS-BD has been reported [21]. We believe that this concept is also useful for ERCP, and thus, we have established a novel transpapillary biliary PS replacement method, the snare-over-the-guidewire (SOG) method, and accumulated cases for analysis.

Currently, there is no clear rule as to which type of PS (straight or pigtail) should be used, and this is often determined by the preference of the endoscopist or the institution. However, if the SOG method proves more useful than the conventional approach to the bile duct from the side of the stent (SOS method), the shape of the PS may have a significant impact on the procedure for PS placement with endoscopic re-intervention in mind. Therefore, in this study, we compared the usefulness of the two methods: the SOG method and the SOS method.

## 2. Materials and Methods

### 2.1. Ethical Considerations

The study protocol was approved by the Institutional Review Board of Kindai University (IRB No. R02-272). The study followed the provisions of the Declaration of Helsinki, as revised in Fortaleza, Brazil, in 2013. All authors have full access to all the data in this study and accept the responsibility for submission for publication. Written informed consent was obtained from each patient before the procedure.

### 2.2. Patients

This was a single-center retrospective study conducted at the Kindai University Faculty of Medicine between January 2018 and July 2020. A total of 244 patients who underwent ERCP for biliary PS replacement were enrolled in the study.

Medical records were examined, and information was collected regarding the demographics (age and sex), anatomy of the digestive tract, history of papillary procedures (endoscopic papillotomy (EST), endoscopic papillary balloon dilation, endoscopic papillectomy), degree of oral protrusion (long or short), presence of parapapillary diverticulum, shape of PS to be replaced (straight or not straight), location of bile duct stricture (distal or proximal), reason for stent replacement, experience of endoscopists (experienced or trainee), and presence of adverse events (bleeding, gastrointestinal perforation, postoperative pancreatitis).

### 2.3. ERCP

All ERCP procedures were performed under conscious sedation using a combination of intravenous propofol and pethidine. In our practice, a fellow/resident or an early-career level endoscopist (trainee) performed ERCP under the supervision of experienced endoscopists who performed 300 or more ERCPs annually for at least 5 years (experienced). A duodenoscope (TJF260V, Olympus Medical Systems, Tokyo, Japan) was advanced into the duodenum, and transpapillary procedures, including cholangiopancreatography and biliary drainage, were performed. Fluoroscopy was used only to produce live imaging and essential static images, while other procedures were performed with reference to endoscopic or ultrasound images.

### 2.4. Endoscopic Replacement of Transpapillary Biliary PS

The SOS method was used as a conventional method for endoscopic replacement of the transpapillary biliary PS. In this method, biliary cannulation using a 0.025-inch guidewire (GW) (Endoselector, PIOLAX MEDICAL DEVICES, INC Kanagawa, Japan or Visiglide2, Olympus Medical Systems, Tokyo, Japan) was performed from the side of the stent. Then, a GW was inserted beyond the stenosis into the drainage area, and the PS was removed with the use of grasping forceps (Figure 1).

The SOG method was used as a novel method. In this method, a GW was inserted transiently into the drainage area through the lumen of the PS, and the PS was removed using a snare inserted through the GW (Figure 2). If the scope has a wire-lock system function, it can be used to ensure the removal of the PS regardless of whether the guidewire is short or long.

The SOS method can be performed with both straight and double pigtail stents, but the SOG method can only be performed with straight stents. Therefore, if the PS was pigtail type, the SOS method was selected, and if it was straight type, either the SOS or the SOG method was selected at the discretion of the endoscopist.

### 2.5. Outcome Definitions

The primary endpoint was the comparison of the procedural success rates between the SOG and SOS methods. In this study, a “successful procedure” was defined as that in which the PS could be removed, leaving a successfully placed GW in the initially drained area. This comparative evaluation was performed in the following categories: overall, distal bile duct stenosis, and proximal bile duct stenosis.

The secondary endpoints were to compare the time required for PS removal and adverse event (AE) incidence rates and to examine the predictors of unsuccessful procedures. At our hospital, in all ERCP procedures, both the endoscopic and fluoroscopic imaging were video recorded from the beginning to the end of the procedure, enabling detailed examination after the procedure (Endo-and-fluoroscopic recorder, Hitachi, Japan). The time required for PS removal was calculated for each patient by evaluating the recorded videos, defined as the time from which the catheter was first inserted into the duodenum through the scope to the time the PS was removed.

A subgroup analysis was performed in cases where the shape of the PS to be replaced was limited to the straight type to eliminate bias due to the type of PS.

### 2.6. Statistical Analysis

All statistical analyses were conducted using JMP version 12.2 (SAS Institute, Cary, NC, USA). In both the SOG and SOS groups, the success rate of the procedure and the AE incidence rate were evaluated using percentages. The time required for PS removal was evaluated using medians.

Normally distributed variables were compared using Student’s t test, and non-normally distributed variables were compared using the Wilcoxon rank sum test. The factors for which the *p*-value was <0.05 in the univariate analysis were subjected to multivariate analysis using a multiple logistic regression model. Statistical significance was set at *p* < 0.05.

## 3. Results

### 3.1. Baseline Characteristics

The baseline characteristics of all patients who underwent procedures using the SOG and SOS methods during the study period are shown in Table 1. The median age (range) was 76 (30–96) years in the SOG group (40 men and 21 women) and 75 (39–96) years in the SOS group (114 men and 69 women). There were no differences in the anatomy of the digestive tract, history of papillary procedures, degree of oral protrusion, presence of parapapillary diverticulum, location of bile duct stricture, disease, reason for stent replacement, or experience of endoscopists between both groups.

Significant differences were found only for the types of stents. In the SOS group, the percentage of straight-type PSs was 30.0%, while in the SOG group, it was 100.0% (*p* < 0.001)

### 3.2. Comparison of Procedural Success Rates between the SOG and SOS Groups

Table 2 shows the results of the comparison of procedural success rates between the SOG and SOS groups. The procedural success rate in the SOG group was significantly higher than that in the SOS group (90.2% vs. 77.1%, *p =* 0.026).

In terms of the bile duct stenosis site, a significant difference was observed in the proximal biliary stenosis lesion (86.4% vs. 57.1%, *p =* 0.025), but not in the distal biliary stenosis lesion (92.3% vs. 77.1%, *p =* 0.206).

Table 3 shows the results of comparing the procedure success rates between the SOG and SOS groups, only in cases where the PS to be replaced was straight.

There was no significant difference between the two groups in the examination of only the straight-type PS as a replacement. However, in terms of the bile duct stenosis site, although no significant difference was observed, the success rate of the procedure tended to be higher in the SOG group than in the SOS group in the proximal biliary stenosis lesion (86.4% vs. 55.6%, *p =* 0.150).

### 3.3. Comparison of the Time Required for PS Removal and Adverse Event Incidence Rate between the SOG and SOS Groups

Table 4 shows the comparison of the time required for PS removal and the incidence rate of AEs between the SOG and SOS groups.

The time required for PS removal (seconds) was significantly shorter in the SOG group than in the SOS group (306 s vs. 375 s, *p =* 0.012).

There was no difference in AE incidence between the two groups.

### 3.4. Univariate and Multivariate Analyses of Predictors for Unsuccessful PS Replacement

Univariate analysis showed that the SOS method and the presence of proximal biliary stenosis and parapapillary diverticulum were predictive factors for unsuccessful PS replacement. Multivariate analysis also showed that the SOS method (odds ratio (OR) 3.64, 95% confidence interval (CI) 1.48–10.46, *p =* 0.0038) and the presence of proximal biliary stenosis (OR 5.08, 95% CI 2.37–11.21, *p* < 0.0001) and parapapillary diverticulum (OR 4.57, 95% CI 1.91–10.93, *p =* 0.0007) were predictive factors for unsuccessful PS replacement (Table 5).

## 4. Discussion

In this study, we found that the SOG method had a higher stent success rate and shorter procedure time than the SOS method for biliary PS replacement, particularly in cases of proximal hilar bile duct stenosis. In both malignant and benign diseases, the success rate of the procedure was superior in the SOG group (Appendix A.

For successful replacement, a GW must be inserted into the drainage area where PS was inserted. Next, only the PS must be removed, leaving the GW inserted for replacement. However, each procedure may be associated with difficulties in performance.

### 4.1. GW Insertion into the Bile Duct Where the PS Is Inserted

In the conventional method, the GW is inserted through the side of the stent (SOS method). It is well known that anatomical bile duct flexion in the oral protrusion renders biliary cannulation difficult, and it is assumed that the insertion of the PS contributes to the straightening of the bile duct and consequently improves the insertion of the GW. However, in the case of PS placement, the bile duct cavity beside the PS is very narrow, making it physically difficult for the GW to pass through. Repeated GW and catheter approaches to the papilla may result in edematous changes in the papilla, increasing the risk of post-ERCP pancreatitis with prolonged procedure time (Figure 3a,b) [22].

This situation is even more serious in patients with bile duct stenosis. The space beside the inserted PS is narrower than that of conventional bile duct stenosis, and the inability of the GW to pass through is often encountered. This problem is particularly severe in cases of proximal hilar bile duct stenosis. If the site of the biliary stenosis is in the distal bile duct, there is only one direction in which the GW must be inserted; however, if the site of the stenosis is at the proximal hilar bile duct, there are multiple directions in which the GW may be inserted. In many cases, the PS itself blocks the insertion of the GW, causing the GW to be directed to another bile duct that is not in the area to be drained. The greater the degree of bile duct stenosis, the greater the frequency of GW insertion failure (Figure 3c,d).

These results indicate that it is more difficult to insert the GW using the SOS method in proximal bile duct stenosis than in distal bile duct stenosis. These may be the reasons proximal hilar bile duct stenosis was found as a predictor of PS replacement failure in the multivariate analysis.

In addition, it is often experienced that the common bile stones interfere with guidewire manipulation.

If the GW does not pass through the side of the PS, the PS is removed; however, this does not guarantee successful GW insertion. Unsuccessful GW insertion leads to PS replacement failure, which has a significant impact on the patient.

In both malignant and benign diseases, the success rate of the procedure was superior in the SOG group (Appendix A).

The reason for the low success rate of the SOS technique in benign disease was thought to be that the common bile duct stones interfere with guidewire manipulation in the SOS technique.

On the other hand, the SOG technique allows instantaneous GW insertion from the duodenal lumen to the drainage area because this technique uses the PS lumen not only as a drainage route but also as an access route for GWs. In addition, the SOG technique does not cause papillary edema because it does not touch the papilla during GW insertion.

### 4.2. PS Removal While Leaving the GW

In the SOS method, the PS is removed using grasping forceps, but sometimes, the GW interferes with grasping. In instances where the PS is a pigtail, it is necessary to maintain a small distance from the papilla, and simultaneously, the GW may come off.

In the multivariate analysis of this study, one of the predictors of PS replacement failure was the presence of a parapapillary diverticulum. This is thought to be attributed to the parapapillary diverticulum, making it difficult to grasp the PS because the axes of the PS and GW are different from those of the normal one. In addition, there is an instance where the GW is grasped together with the PS, and both come out together. Furthermore, when removing the pigtail PS, the pigtail loop of the PS may become entangled in the GW, and both may become loose. If both the PS and GW are dislodged, bile duct intubation and manipulation of the GW to the drainage branch beyond the stenosis are required, which could be more difficult in cases with serious stenosis. If the case is complicated by cholangitis due to stent occlusion, it may result in a fatal condition.

On the other hand, in the SOG method, the PS is grasped by the snare forceps inserted through the GW. Because the PS and GW axes are coaxial, PS grasping is very quick and easy. In gripping with snare forceps, only the PS is gripped, not the GW, so the PS can be easily removed while leaving the GW. In this study, there were no cases in which the GW was lost along with the PS.

In the analysis in terms of the site of stenosis, the usefulness of the SOG method over the SOS method was significant in cases of proximal bile duct stenosis, and the SOG method is recommended in cases of proximal bile duct stenosis where PS replacement is difficult.

The SOG method has a limitation in that it can only be performed with a straight PS. In contrast, the SOS method can be performed for both straight and pigtail types. If the stents to be removed are pigtail stents, the SOG method cannot be performed due to its nature. Pigtail stents, for which the SOG method cannot be used, were assumed to be a predictor of unsuccessful stent replacement but were not significant in univariate analysis.

One reason is that both the SOG method and the SOS method were used to remove straight stents, so it is considered that the existence of cases in which straight stents were difficult to replace had an effect.

However, a comparison of the success rates of the two techniques in cases confined to straight PS also showed that the SOG method was superior to the SOS method, at 90.2% and 76.4%, respectively, although the difference was not significant. Even in a study limited to the straight type, the superiority of the SOG method tended to be more pronounced in proximal drainage than in distal drainage.

In cases with a long stenosis such as hilar stenosis, a pigtail stent may be required because the straight stent is not long enough. The pigtail stent is still an important device in clinical practice, and it is expected that the pigtail stent will be developed with re-intervention in mind in the future.

### 4.3. Limitations

This study has some limitations. First, this was a single-center retrospective study with a limited number of patients. Second, this study was designed to compare the methods of stent replacement and did not evaluate the patency rate or deviation of reimplanted stents.

However, despite these limitations, the results of the present study suggest that straight PS should be selected for proximal bile duct drainage, which will have a significant impact on actual clinical practice. As unsuccessful re-intervention is very burdensome to the patient, the PS selection strategy should take re-intervention into account.

## 5. Conclusions

The SOG method may be a very useful procedure for PS placement with possible re-intervention. A prospective comparison study using only a straight PS is required in the future to confirm these findings.

## Figures and Tables

**Figure 1 jcm-10-02858-f001:**
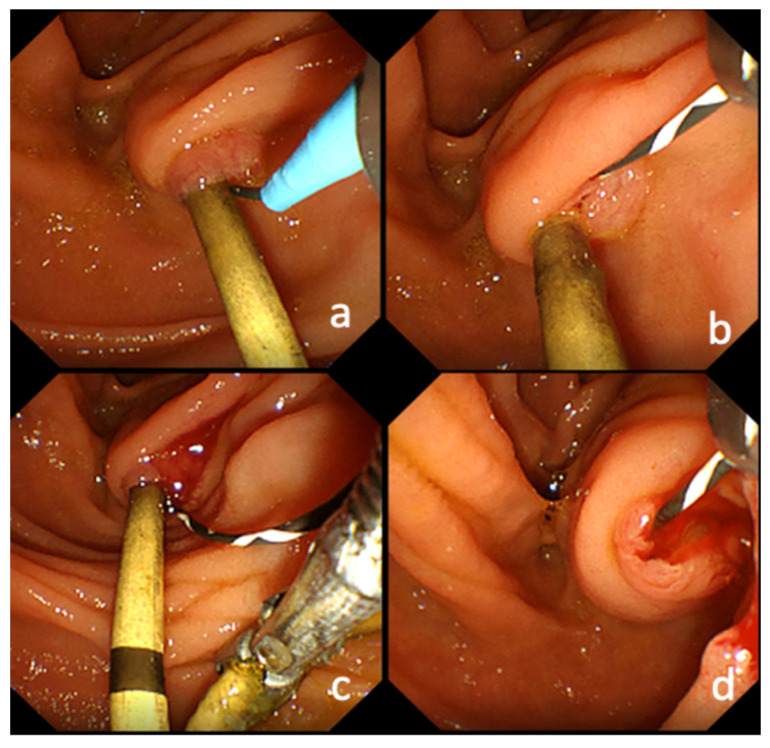
Endoscopic image of the side-of-stent method. (**a**). Biliary cannulation using a guidewire was performed from the side of the stent. (**b**). A guidewire on the side of the stent is visible. (**c**). Scratching through the guidewire and gripping the stent with gripping forceps. (**d**). The PS was removed with grasping forceps while leaving the guidewire.

**Figure 2 jcm-10-02858-f002:**
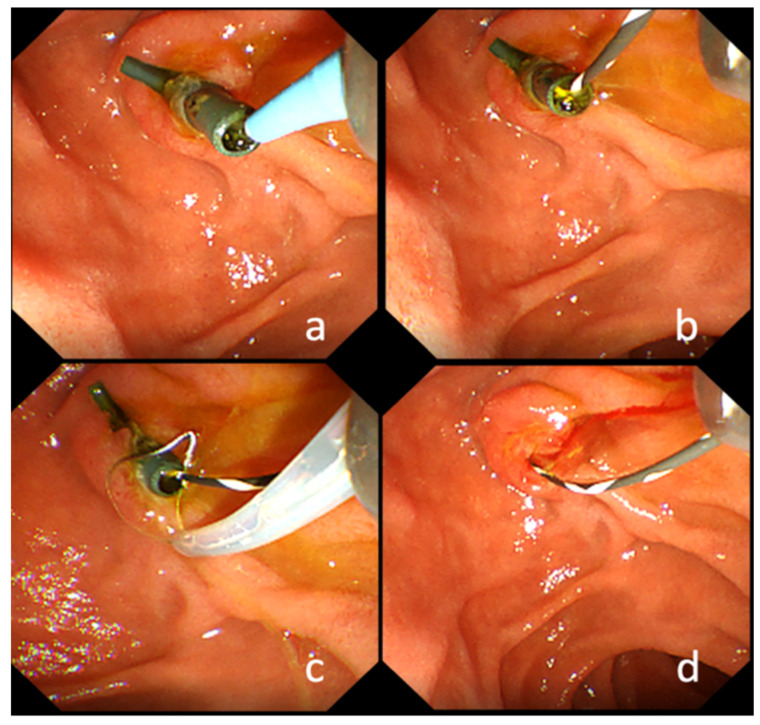
Endoscopic image of the snare-over-the-guidewire method. (**a**). A guidewire was inserted transiently into the drainage area through the lumen of the PS. (**b**). A guidewire passing through the stent lumen is visible. (**c**). The PS was grasped using a snare inserted through the guidewire. (**d**). The PS was removed using a snare while leaving the guidewire.

**Figure 3 jcm-10-02858-f003:**
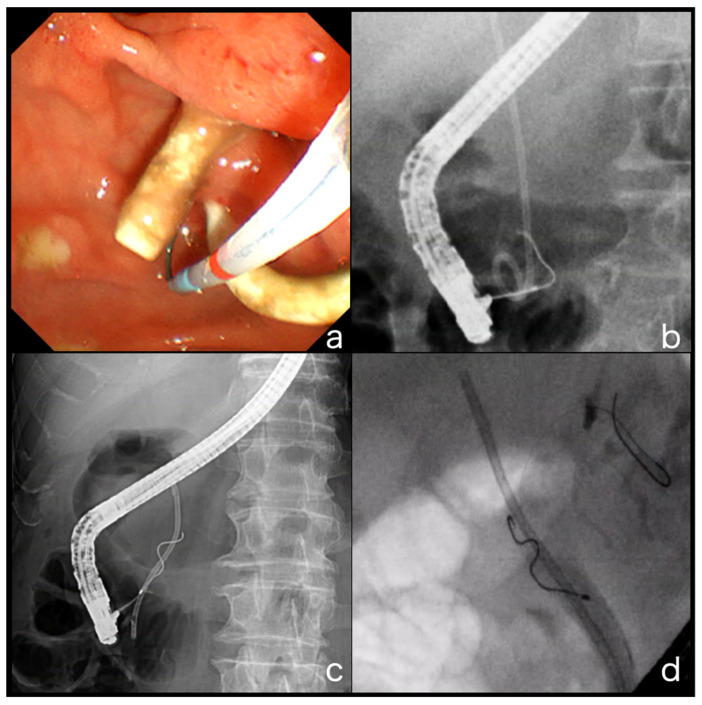
The difficulty of inserting the GW into the bile duct where the PS is inserted. (**a**,**b**) In the case where a PS is placed, the bile duct cavity beside the PS is very narrow, making it physically difficult for the GW to pass through. (**c**,**d**) The situation is even more serious in the presence of bile duct stenosis. The space beside the inserted PS is narrower than that of the conventional bile duct stenosis, and instances in which the GW cannot pass through are often encountered.

**Table 1 jcm-10-02858-t001:** Baseline characteristics.

	SOG Group (*n* = 61)	SOS Group (*n* = 183)	*p*-Value
Age (median, range)	76 (30–96)	75 (39–96)	0.83
Female (*n*, %)	21(34.4)	69 (37.7)	0.76
Anatomy of the digestive tract (*n*, %)			0.43
Normal	54 (88.5)	169 (92.3)	
Postoperative	7 (11.5)	14 (7.7)	
History of papillary procedures (*n*, %)			1.00
Existed	42 (68.9)	126 (68.9)	
None	19 (31.1)	57 (31.1)	
Degree of oral protrusion (*n*, %)			0.62
Long	7 (11.5)	17 (9.3)	
Short	54 (88.5)	166 (90.7)	
Parapapillary diverticulum (*n*, %)			0.68
Existed	8 (13.1)	30 (16.4)	
None	53 (86.9)	153 (83.6)	
Stent type (*n*, %)			<0.0001
Straight	61 (100.0)	55 (30.0)	
Not straight	0 (0.0)	128 (70.0)	
Location of bile duct stricture (*n*, %)			0.06
Distal	39 (63.9)	141 (77.0)	
Proximal	22 (36.1)	42 (33.0)	
Disease (*n*, %)			
Malignant	31 (51%)	82 (45%)	
Cholangiocarcinoma	16 (26%)	34 (19%)	
Pancreatic cancer	10 (16%)	37 (20%)	
Others	5 (9%)	11 (6%)	
Benign	30 (49%)	101 (55%)	
Bile duct stone	22 (36%)	71(39%)	
Others	8 (13%)	30 (16%)	
Reasons for stent replacement (*n*, %)			0.21
Stent occlusion	16 (26.2)	65 (35.5)	
Regular exchange	45 (73.8)	118 (64.5)	
Operator (*n*, %)			0.88
Experienced	19 (31.1)	60 (32.8)	
Trainee	42 (68.9)	123 (67.2)	

*p* < 0.05 was considered statistically significant. SOG: snare-over-the-guidewire method, SOS: side-of-stent method, IQR: interquartile range.

**Table 2 jcm-10-02858-t002:** Comparison of procedural success rates between the SOG and SOS groups.

	SOG Group	SOS Group	*p*-Value
	Overall		
	(*n* = 61)	(*n* = 183)	
Success rate of removing the stent after inserting the guidewire into the bile duct where the stent was placed (%)	90.2 (55/61)	77.1 (141/183)	0.026
	Distal		
	(*n* = 39)	(*n* = 141)	
Success rate of removing the stent after inserting the guidewire into the bile duct where the stent was placed (%)	92.3 (36/39)	77.1 (117/141)	0.206
	Proximal		
	(*n* = 22)	(*n* = 42)	
Success rate of removing the stent after inserting the guidewire into the bile duct where the stent was placed (%)	86.4 (19/22)	57.1 (24/42)	0.025

*p* < 0.05 was considered statistically significant. SOG, snare-over-the-guidewire method; SOS, side-of-stent method; IQR, interquartile range.

**Table 3 jcm-10-02858-t003:** Comparison of procedural success rates between the two groups (only in cases where the PS to be replaced was straight).

	SOG Group	SOS Group	*p*-Value
	Overall		
	(*n* = 61)	(*n* = 55)	
Success rate of removing the stent after inserting the guidewire into the bile duct where the stent was placed (%)	90.2 (55/61)	76.4 (42/55)	0.077
	Distal		
	(*n* = 39)	(*n* = 46)	
Success rate of removing the stent after inserting the guidewire into the bile duct where the stent was placed (%)	92.3 (36/39)	80.4 (37/46)	0.210
	Proximal		
	(*n* = 22)	*(n* = 9)	
Success rate of removing the stent after inserting the guidewire into the bile duct where the stent was placed (%)	86.4 (19/22)	55.6 (5/9)	0.150

*p* < 0.05 was considered statistically significant. SOG, snare-over-the-guidewire method; SOS, side-of-stent method; IQR, interquartile range.

**Table 4 jcm-10-02858-t004:** Comparison of the time required for PS removal and adverse event incidence rate between the SOG and SOS groups.

	SOG Group	SOS Group	*p*-Value
	(*n* = 61)	(*n* = 183)	
Time required to insert the guidewire into the bile duct where the stent was placed and remove the stent (Successful cases only) (median; seconds (IQR))	306 (230–400)	375 (268–573.5)	0.012
Adverse event incidence rate (%)	0 (0/61)	0.55 (1/183)	1.00
post-procedure pancreatitis	0	1	
bleeding	0	0	
perforation	0	0	

*p* < 0.05 was considered statistically significant. PS, plastic stent; SOG, snare-over-the-guidewire method; SOS, side-of-stent method; IQR, interquartile range.

**Table 5 jcm-10-02858-t005:** Univariate and multivariate analyses of predictors for unsuccessful PS replacement.

	Univariate Analysis			Multivariate Analysis		
	OR	95% CI	*p*-Value	OR	95% CI	*p*-Value
SOS method	2.73	1.10–6.79	0.026	3.64	1.48–10.46	0.0038
Stent type			0.2598			
Proximal biliary stenosis	2.77	1.43–5.37	0.0032	5.08	2.37–11.21	<0.0001
Postoperative anatomy	2.22	0.85–5.85	0.15			
Stent occlusion	1.41	0.74–2.72	0.31			
Trainee	1.65	0.76–3.58	0.40			
Naïve papilla	0.89	0.45–1.78	0.86			
Long oral protrusion	2.25	0.90–5.62	0.10			
Parapapillary diverticulum	2.54	1.19–5.45	0.0243	4.57	1.91–10.93	0.0007

*p* < 0.05 was considered statistically significant. OR, odds ratio; CI, confidence interval; SOS, side-of-stent method; IQR, interquartile range.

## Data Availability

All the data used for this analysis can be confirmed at any time.

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
