# Peer review of "Usefulness of the Novel Snare-over-the-Guidewire Method for Transpapillary Plastic Stent Replacement (with Video)"

_jcm, 2021, doi:10.3390/jcm10132858_

Round 1

Reviewer 1 Report

This is a well designed, well described paper on the difficulty of attaining wire access for biliary strictures when a stent exchange is needed. The authors describe well that using a snare over guide wire is a simple addition to the ERCP procedure with great results in clinical success. 

Comments:

  • In baseline characteristics there exists a significant difference in stent type, with no pigtail stents used in the SOG group, which I understand because a pigtail stent does not allow the SOG technique due to its nature. However in the univariate analysis it fails to attain an impact on the success of the procedure between the techniques. I find that this is an important point that is not mentioned again in the discussion of the paper.
  • For the generalizability of the paper to our audience, can you comment if this technique can only be used in short wire exchange system or may be used in both a short or long wire system. In past experience SOG technique is not feasible in long-wire as there is no lock on the wire, but perhaps the authors have different experience. 
  • Introduction that PS stent may be used for benign or malignant stenosis. In the discussion the authors mainly describe proximal stenosis as malignant hilar strictures. Do these results also pretain to benign strictures. What percentage of your results were from benign strictures? Please elaborate. 

Author Response

Reviewer 1

Comments:

In baseline characteristics there exists a significant difference in stent type, with no pigtail stents used in the SOG group, which I understand because a pigtail stent does not allow the SOG technique due to its nature. However, in the univariate analysis it fails to attain an impact on the success of the procedure between the techniques. I find that this is an important point that is not mentioned again in the discussion of the paper.

Thank you for the important point out.

We have added the following to the discussion part in response to your suggestion.

 If the stents to be removed are pigtail stents, the SOG method cannot be performed due to its nature. Pigtail stents, for which the SOG method cannot be used, were assumed to be a predictor of unsuccessful stent replacement but were not significant in univariate analysis.

One reason is that both the SOG method and the SOS method were used to remove straight stents, so it is considered that the existence of cases in which straight stents were difficult to replace had an effect.

 In cases with a long stenosis such as hilar stenosis, a pigtail stent may be required because the straight stent is not long enough. The pigtail stent is still an important device in clinical practice, and it is expected that the pigtail stent will be developed with re-intervention in mind in the future.

For the generalizability of the paper to our audience, can you comment if this technique can only be used in short wire exchange system or may be used in both a short or long wire system. In past experience SOG technique is not feasible in long-wire as there is no lock on the wire, but perhaps the authors have different experience. 

Thank you for your important remarks.

Even if the guidewire is long, the SOG method can definitely remove the PS with the guidewire intact by using the wire lock system of the scope.

We have added this point to the explanation part of the SOG method as follows.

“If the scope has a wire-lock system function, it can be used to ensure the removal of the PS whether the guidewire is short or long.”

Introduction that PS stent may be used for benign or malignant stenosis. In the discussion the authors mainly describe proximal stenosis as malignant hilar strictures. Do these results also pretain to benign strictures.

What percentage of your results were from benign strictures? Please elaborate. 

Thanks for pointing this out.

We have added the breakdown of Diseases to Table 1, based on your suggestion.

We have also added the following to the Results and Discussion part.

 In both malignant and benign diseases, the success rate of the procedure was superior in the SOG group. (Supplemental Table 1)

 In addition, it is often experienced that the stones interfere with guidewire manipulation.

Reviewer 2 Report

Main Comment:

In this manuscript two techniques for transpapillary plastic stent replacement are compared (snare-over-the-guidewire verus side-of-stent method). This is a single-center retrospective evaluation with a limited number of patients; however, it may be seen as an incentive to further studies.

Additional Comments:

Figures 1 and 3 do not yield additional information and can be omitted (the two techniques are well illustrated by Figures 2 and 4).

Table 4: The adverse event in the SOS group should be specified.

Discussion, line 251-252: "For successful biliary PS replacement, a GW must first be inserted into the bile duct that is drained by PS removal." - This sentence is not clear ("drained by PS removal"?), please rephrase it.

Author Response

Reviewer 2

Figures 1 and 3 do not yield additional information and can be omitted (the two techniques are well illustrated by Figures 2 and 4).

Thank you for advice.

We omit Figure 1 and 3 as your suggestion.

We are confident that we are able to demonstrate the actual technique in the video.

Table 4: The adverse event in the SOS group should be specified.

Thank you for pointing this out.

This complication was postoperative pancreatitis and I have added a correction to the Table for clarity.

Please confirm.

Discussion, line 251-252: "For successful biliary PS replacement, a GW must first be inserted into the bile duct that is drained by PS removal." - This sentence is not clear ("drained by PS removal"?), please rephrase it.

Thank you for suggestion.

We revised the sentence to For successful replacement, a GW must be inserted into the drainage area where PS was inserted."

This revision has improved what we wanted to convey in a clearer way.

Thank you very much.

Round 2

Reviewer 1 Report

The authors have adequately addressed the prior concerns well.